# Instagram-Based Benchmark Dataset for Cyberbullying Detection in Arabic Text

Reem ALBayari [1,2,*] and Sherief Abdallah [2]

1   Higher College of Technology, Abu Dhabi P.O. Box 25026, United Arab Emirates
2   Faculty of Engineering and IT, The British University in Dubai, Dubai P.O. Box 345015, United Arab Emirates; sherief.abdallah@buid.ac.ae
*   Correspondence: ralbayari@hct.ac.ae; Tel.: +971-556780224

**Abstract:** (1) Background: the ability to use social media to communicate without revealing one's real identity has created an attractive setting for cyberbullying. Several studies targeted social media to collect their datasets with the aim of automatically detecting offensive language. However, the majority of the datasets were in English, not in Arabic. Even the few Arabic datasets that were collected, none focused on Instagram despite being a major social media platform in the Arab world. (2) Methods: we use the official Instagram APIs to collect our dataset. To consider the dataset as a benchmark, we use SPSS (Kappa statistic) to evaluate the inter-annotator agreement (IAA), as well as examine and evaluate the performance of various learning models (LR, SVM, RFC, and MNB). (3) Results: in this research, we present the first Instagram Arabic corpus (sub-class categorization (multi-class)) focusing on cyberbullying. The dataset is primarily designed for the purpose of detecting offensive language in texts. We end up with 200,000 comments, of which 46,898 comments were annotated by three human annotators. The results show that the SVM classifier outperforms the other classifiers, with an F1 score of 69% for bullying comments and 85 percent for positive comments.

**Keywords:** cyberbullying; offensive language; Arabic dialect

## 1. Introduction

Throughout the recent years, the number of social media users has grown dramatically. Facebook, Twitter, Instagram, and many more platforms provide a perfect place for users to express their thoughts and interact with other users from different cultures and backgrounds. Unfortunately, this enriching social experience also provides a fertile environment for cyberbullying [1,2]. Cyberbullying is defined as the use of telecommunications to disseminate abusive behavior such as messages, images, or videos with the aim of causing harm to others [3]. Such a toxic environment can cause hate crimes and psychological harm [4]. This provoked the necessity for automatic detection of offensive and abusive speech over social media platforms. Due to the negative effects of cyberbullying [5,6], the recent years have seen an increasing number of studies that collected and annotated datasets related to cyberbullying from different social media platforms [7–9]. Since the social media data comes from the users' minds, it provides an unprecedented opportunity for studying cognitive processes such as perception, personality, and information spread [10]. For instance, the authors in [11] conducted an online survey and discovered the granular functional impact of social media in supporting a positive impression of stresses throughout the pandemic. As a result of collective resilience, this study provides an empirically verified theoretical framework for understanding the emergence of social media buffering mechanisms.

In addition to that, social media presents challenges since it necessitates the development of appropriate interpretable frameworks that can highlight the structure of knowledge that flows through social connections. Indeed, the science of complex networks has recently progressed beyond purely sociological analysis to include the cognitive aspects of the social



media [12]. Utilizing cognitive networks to connect big data analytics with interpretable modeling will considerably help numerous cognition research fields [10]. Therefore, in this work, we contribute to minimizing the gap in the field of Arabic language modeling through introducing the first Instagram Arabic corpus (sub-class categorization (multi-class)). We focus on the Instagram platform because it represents the third most used platform in terms of the number of active users [13]. More importantly, Instagram is popular amongst teens, who are particularly vulnerable to cyberbullying [14]. Furthermore, Instagram does not restrict users from posting objectionable content, and if such content is posted, it can take a long time to be removed [15]. This corpus can be utilized for multiple purposes, such as cyberbullying auto-detection or dialect identification or sentiment analysis.

In the following section, we provide a quick overview of Arabic and go through the linguistic characteristics that make it one of the most difficult languages to classify, and of the most recent Arabic datasets that addressed cyberbullying and abusive, offensive language, including dataset size, dialect, collection and annotation method, and dataset source. Then in Section 3, we describe the research methodology; dataset collection and preprocessing, labeling method, annotation schema, and descriptive analysis, as well as dataset evaluation, followed by Section 4's results and analysis. In the last section, we conclude and discuss the limitations and future work.

## 2. Related Work

Arabic is one of the world's six most widely spoken languages. Nearly 300 million people speak Arabic as their first language [16]. Arabic contains linguistic features that make it one of the most difficult languages to speak in comparison to other languages. Furthermore, its words do not have sequential forms because their word structures vary depending on where they appear in the phrase and what they imply [16,17]. These changes, along with the complete lack of any structural rules, make processing this language extremely difficult [18]. Thus, the need for an existing Arabic dataset to evaluate automatic algorithms for detecting and classifying offensive speech is critical. Indeed, the majority of the available cyberbullying datasets are in English [19–21], while there are only a few available in Arabic. Hence, in this section, we provide an overview of the most recent Arabic datasets that addressed cyberbullying, abusive, and offensive language. For abusive language detection, Mubarak et al. in [16] extracted a dataset from an Arabic news channel (Aljazeera.net) using Twitter streaming API. Their dataset consists of 32 K comments. They classified the comments manually into three categories (533 obscene, 25,506 offensive, and 5653 clean). The authors did not include the positive class, and also the dataset is unbalanced. In our work, though, we included the three categories (positive, negative, and neutral). We focused mostly on the negatives to make the dataset multipurpose, such as in the use of sentimental analysis. It includes Modern Standard Arabic (MSA) and various dialects. They made it public. Haidar et al. in [22] collected a dataset from Twitter. The authors manually labeled 34,890 Arabic tweets under two categories, bullying, and non-bullying: 2999 and 31,891, respectively. Various dialects are included in the dataset (from Lebanon, Egypt, and the Gulf area). It is not public. The authors in [18] collected a dataset (15,050 Arabic comments) from YouTube for detecting anti-social behavior in online communications. Three annotators from three different Arab countries classified the dataset into positives and negatives. There were 3532 and 11,518, respectively. Variant dialects are included in the dataset. Iraqi, Egyptian, and Libyan nationals make up the majority of them. The authors made the dataset publicly available. Otiefy et al. in [23] gathered balanced dataset messages between two participants. The authors manually annotated 3200 messages, 1600 offensive, and 1600 non-offensive messages, in total. The authors collected messages that contain vulgar language, sentences that try to harshly criticize or mock someone, and obscene and offensive expressions. The authors in [24] created a dataset for detecting abusive/hate speech in a single dialect: the Levantine dialect (Syrian and Lebanese) from Twitter based on several queries formed from possible targets of abusive/hate speech. The authors manually annotated 6000 tweets under three categories

(3650 normal, 1728 abusive, and 468 hate). A small and unbalanced dataset may affect the classifier's performance. (Al-Ajlan and Ykhlef, 2018) in [25] collected from Twitter 20 k tweets for cyberbullying detection utilizing deep learning. The authors classified the tweets into two categories (bullying/nonbullying). (O. Hosam, 2018) in [26] collected 35,098 posts and tweets from Facebook and Twitter. The authors classified the posts under six categories (15,294 toxic, 1595 severe toxic, 8449 obscene, 478 threats, 7877 insults, and 1405 identity hate). The author labeled 10% of the dataset manually and the remaining 90% was based on the presence of specific terms, which are referred to as the "poison keywords list". Automatic labeling based on a specified list could provide unreliable labeling, particularly in the case of the Arabic dialect [27]. In addition, the total number of labeled comments for each category differs significantly (unbalanced dataset).

According to the above references, most of the work focused on the task of binary classification, i.e., they are labeled as (bullying/non-bullying) or (offensive/non-offensive) [18,22,23,25]. Nonetheless, multi-class integration is becoming an increasingly important [28]. That is because it is not used for a specific classification. Therefore, in our work, we introduce a corpus with (sub-class categorization (multi-class)) that could help in studying cognitive processes, such as investigating whether cyberbullying is linked with increased levels of anxiety in online audiences. However, the majority of the available cyberbullying datasets are in English [19,20], with only a few available in Arabic, none of which are data collected from the Instagram platform [18,21]; the number of Arabic datasets available for offensive/hate speech/cyberbullying auto detection is limited compared to in the English language. Furthermore, none of them are collected from Instagram. While for other languages, Instagram was used by researchers with the aim of autodetecting cyberbullying. For instance, the authors in [7] collected 900 Turkish-language comments from Instagram and Twitter, 450 of which had cyberbullying content and 450 of which did not. The scraping tool employed by the authors was not mentioned. The authors examined the performance by utilizing four classical classifiers (SVM, NBM, J48, and J48) and their accuracy results, respectively, were (64%, 81%, 54%, and 81%). The authors in [15] collected 3000 images from public accounts of Instagram through the site's official API. This is to develop early-warning systems for recognizing images that are vulnerable to attacks. The authors in [29] collected 697 K media sessions from Instagram consisting of images and their associated comments. The authors designed a labeling study for cyberbullying, as well as image content using human labelers. In addition, they analyzed the labeled data and presented the correlations of the different features between cyberbullying and cyberaggression.

## 3. Methods

In this section, we describe the research methodology, dataset collection, and preprocessing, labeling annotation evaluation, as well as dataset evaluation.

### 3.1. Dataset Collection and Preprocessing

Table 1 shows examples of offensive language with the translation that can be used in the digital platforms. When we used Google Translate to translate the comments from Arabic to English, the translations were inaccurate; for instance, the word "وحش" was translated to "beast" when in reality it means "hideous". That is why we used the help of native speakers. This proves that working in the Arabic dialect is more complicated than in English since the words do not get translated as accurately as when English words are translated.

**Table 1.** An instance of offensive language in digital platforms.

| Comments Translation | Comments |
|---|---|
| Whore. | يا فاجره |
| Why ok? This is low level, madam. | ليه طيب؟ ده رخص يا مدام |
| I love you, but why are you so raffish? | انا بحبك بس انتي بقيتي سفله ليه كده |
| A hideous and disgusting, don't wear this style because it is not sweet at all. | وحش ومقزز جدا يريت، بلاش الاستايلدة مش حلو فيكي خالص |

To locate accounts, we conducted an internet search for social influencers in the Arab world (including fashionistas, singers, YouTubers, and bloggers) who had been subjected to cyberbullying.

We used Google with the below queries to find out the accounts:

- Arabic fashionistas who had been suffering/subjected to bullying,
- Arabic singers who had been suffering/subjected to bullying,
- Arabic YouTubers who had been suffering/subjected to bullying,
- Arabic bloggers who had been suffering/subjected to bullying.

We used the following quality assessment criteria to ensure that selected accounts would satisfy the aim of this study.

- Instagram profiles,
- Arabic accounts,
- The minimum number of comments on the post is 200.

We crawled Instagram in March 2021 and posted dates ranging from 2019 to 2021. We collected in total 200,000 Arabic comments utilizing the official Instagram APIs. Our dataset includes the following eight attributes: text, date, timestamp, user_id, username, profile_URL, profile_pic_url, and comment_id. We filtered out the non-Arabic text and ended up with 198,000 comments, written in different dialects. Then, we eliminated symbols such as @ and #, digits, and non-Arabic characters, URLs, and user mentions. Figure 1 illustrates the research methodology.

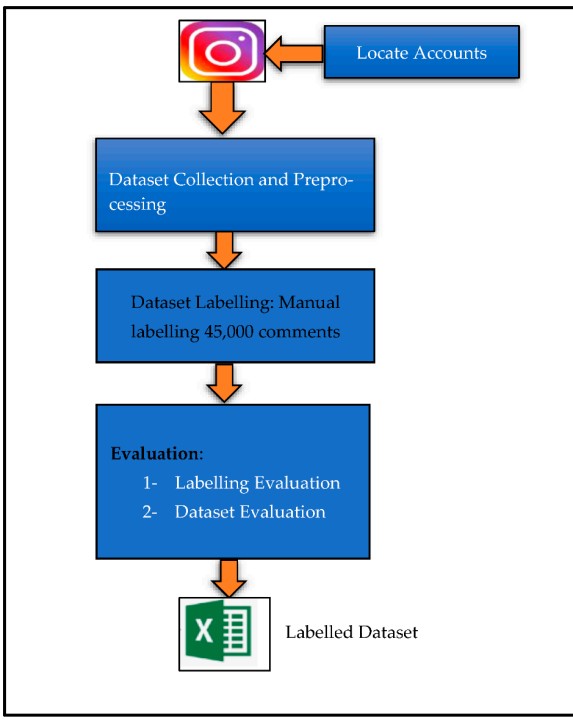

**Figure 1.** Illustration of the research methodology.

*3.2. Dataset Labeling*

We chose the manual labeling approach since auto-labeling based on a predefined list [27] could lead to inaccurate labeling, since there are different dialects, as well as some comments that could hold bullying/hating without having any clear offensive words, such as, for instance, "وين زوجك عنك؟", which means for Arabic people, "Your husband should prevent you from doing such a bad behavior". We considered distinctions in Arab dialects, as well as the diversity of offensive language views across the Arab world. Three annotators representing three different Arabic dialects were assigned to the labeling task (one is Jordanian, one is Egyptian, and one is Iraqi). All the annotators had a bachelor's degree and their ages ranged from 23 to 27 years. We opted for young adults because they have recent and up-to-date experience with cyberbullying. In this work, we created a multi-labeling dataset. To the best of our knowledge, this is the first dataset with multi-labeling for cyberbullying detection in Arabic text. After we classified the comments into (positive/negative/neutral), we did further classification for the negative comments by classifying them according to their level of negativity into two categories (toxic and bullying), as shown in Figure 2. The dialect was also manually labeled by the annotators. If the comment's dialect was not obvious, NA was written by the annotators (not available). In addition, if just emojis are used in the comments, the dialect is NA.

| Labelling | | | |
|---|---|---|---|
| **text** | **Positive/Negative/Neutral** | **Positive/Neutral/Bullying/Toxic** | **Dialect** |
| أد ايش صار عمرا؟ | Neutral | Neutral | Levantine |
| بصراحه منظرك مقزز جدا | Negative | Bullying | Levantine |
| لك شو هذا الجسم القذر. | Negative | Bullying | Levantine |
| ألف مبروووك | Positive | Positive | Levantine |
| الموت قادم لا محاله قد متى ايه معرفش ليه كدا انتي مصره تزعليني.... | Negative | Toxic | Egyptian |

**Figure 2.** Labeling samples.

Annotation Scheme

We provided the three annotators with the guidelines below before they started annotating the dataset.

In the first stage, the annotators were requested to label the comments into three categories (normal, positive, or negative) based on the guidelines below:

- Annotators have been encouraged to avoid interpreting text subjectivity based on their own feelings and other background information in general. This is because the types of sentiment expressed may differ depending on the annotator's or reader's background knowledge [30]. For example, "يجب وآد البنات" means "We have to bury girls alive." If the readers do not like girls, they could find this comment neutral and not offensive.
- Be as consistent as possible in the whole annotation journey.
- Document any questions you may have or issues you may come across and report them back to the data team. These questions will be valuable feedback for further expansion and improvements of this document.
- Negative: if there is no offensive, aggressive, insulting or profanity content. Positive: if it contains any words of praise, thanks, appreciation, etc.
- Normal: anything else, such as announcements, benedictions, etc.

In the second stage, the annotators were requested to relabel the negative comments into two categories (bullying, toxic,) based on the guidelines below:

- Toxic: for the comments that hold bad feelings but do not consist of any bad words.
- Bullying: for comments that include extreme (abusive language/offensive, insulting, and aggressive) language based on some characteristics such as race, color, ethnicity, gender, sexual orientation, nationality, religion, or others. This labeling was chosen along with the hate speech definition in [3] [Nockleby, 2000]. Examples for the labeling are missioned in Table 2.
- If there is a disagreement, we used majority voting.

**Table 2.** Examples for the labeling.

| Comments in English | Comments in Arabic | Classification |
|---|---|---|
| Follow me too | تابعني كمان | Neutral |
| Watch my story | شاهد لستوري | Neutral |
| God forgive me, I did not expect her to do such a thing! | استغفر الله العظيم ، ما توقعت تعمل كذا | Toxic |
| Death is inevitably coming, I don't know why . . . you insist on making me mad | الموت قادم لا محاله قد متى ايه انتي مصره تزعليني . . . معرفش ليه كدا | Toxic |
| He doesn't deserve her, see how she looks like, she is from Hollywood. And he is "No comment" | ...ما يستاهلها شوفوا شكلها كأنها من هوليوود وهو نو كومنت | Toxic |
| Uglier than ugliness | أبشع من البشاعة | Bullying |
| Bitch | ساقطه | Bullying |

### 3.3. Dataset Descriptive Analysis

A total of 46,898 comments were labeled: 18,193 as negative, 17,376 as positive, and 11,329 as neutral. Although imbalanced datasets create a challenge for most learning algorithms [31,32], there are many techniques to overcome this problem.

In the second stage, the annotators were requested to relabel the negative comments into two categories (bullying or toxic) based on the guidelines mentioned in Section 3.3. The final corpus had 12,256 bullying comments, 5937 toxic comments, 17,376 positive comments, and 11,329 neutral comments.

We conducted an analysis to see if the comments were posted or viewed by people from a wide range of Arab nations to learn more about what our dataset represents. This is because the presence of people from various Arab countries in a chat is taken as a sign that the majority recognizes the insults. We calculated and illustrated the number of comments per the dialect in Figure 3. Our dataset consisted of four different dialects: Egyptian, MSA, Gulf, and Levantine. The Egyptian dialect comprised the biggest percentage of the dataset, followed by Gulf and MSA, see Figure 4. However, the percentage of Levantine was the lowest. NA presented 20% of the dataset and we did not consider it as a dialect since it refers to emojis or unknown dialects. However, the presence of many dialects in the dataset makes the cyberbullying classification challenging. This is because there are several dialects and the number of keywords connected with each dialect increases because words in the Arabic language differ from dialect to dialect [31]. This is because when the number of keywords for a certain dialect increase, it decreases the classifier performance.

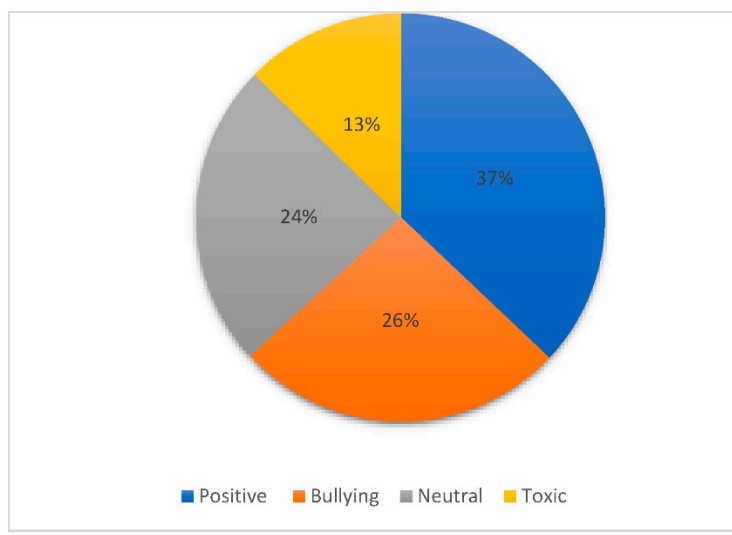

**Figure 3.** Total number of labeled comments per category.

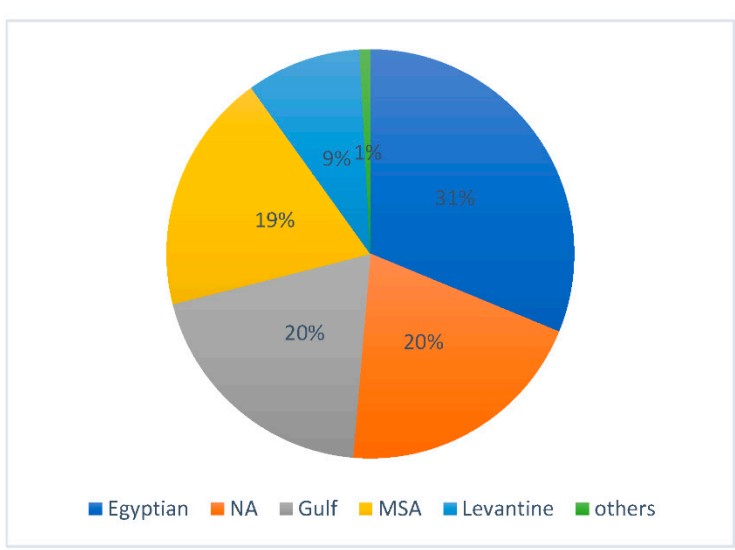

**Figure 4.** Percentage of comments per dialect.

*3.4. Evaluation*

3.4.1. Labeling Evaluation

For a dataset to be regarded as a benchmark, we adopt Fleiss's Kappa metric to measure the inter-annotator agreement (IAA) [31,33–35].

3.4.2. Benchmark Evaluation

To consider the dataset as a benchmark, we examined and evaluated the performance of various learning models. In our baseline experiments, we applied some preprocessing to the data by removing the stop-words, symbols, URLs, and non-Arabic characters. In addition, we applied word tokenization and stemming. We utilized Tf-Idf to extract the text data's features, and then we implemented some of the most common classical classifiers [28,36–38]. For the classical classifiers approach, we split the dataset into 80% training and 20% testing.

- Logistic regression (LR): this is a predictive model. It is a statistical learning technique used for the task of classification. Even though the name of the classifier has the word 'regression' in it, it is used to produce discrete binary outputs.

- Multinomial naïve Bayes (MNB): this classifier estimates the probability of each class label, based on the Bayes theorem, for some texts. The result of this is the class label with the highest probability score. MNB assumes the features are independent and, as a result, all features contribute equally to the computation of the predicted label.
- Support vector machines (SVM): SVM is a very prevalent supervised classifier.
- It is non-probabilistic. SVM uses hyperplanes to segregate labels. SVM supports linear and nonlinear models. Basically, each hyperplane is expressed by the input documents (vector).
- Random forest (RF): RF is a supervised learning-based classifier. This ensemble model utilizes a set of decision trees, which computes the resulting label aggregately.

## 4. Result and Analysis

In this section, we evaluated both the agreement level between the annotators of the labeling and the performance of various baseline learning models. For evaluating the inter-annotator agreement (IAA), we adopt Fleiss's Kappa to measure the agreement between the three annotators. Results show that the total Fleiss Kappa coefficient is = 0.869 with a $p$ value of $10^{-3}$ on 9130 comments, which indicates near-perfect agreement among the three annotators. For the classifiers' performance evaluation, we examined and evaluated the performance of various baseline learning models utilizing four key metrics (accuracy, presession, recall, and F1 score). The results in Tables 3–6 illustrate the classifiers' fusion metrics for each classifier. The section below provides details on the evaluation details and baseline results.

**Table 3.** Logistic regression (LR) Classifier Fusion Metrics.

| Resulting Tags | Precision | Recall | F1-Score |
|---|---|---|---|
| Bullying | 0.70 | 0.58 | 0.63 |
| Neutral | 0.65 | 0.52 | 0.58 |
| Positive | 0.77 | 0.90 | 0.83 |
| Toxic | 0.30 | 0.37 | 0.33 |
| Macro avg | 0.60 | 0.59 | 0.59 |
| Weighted avg | 0.66 | 0.59 | 0.59 |
| Accuracy | | | 0.66 |

**Table 4.** Random Forest (RF) Classifier Fusion Metrics.

| Resulting Tags | Precision | Recall | F1-Score |
|---|---|---|---|
| Bullying | 0.62 | 0.72 | 0.67 |
| Neutral | 0.63 | 0.56 | 0.60 |
| Positive | 0.76 | 0.90 | 0.82 |
| Toxic | 0.42 | 0.13 | 0.20 |
| Macro avg | 0.61 | 0.58 | 0.57 |
| Weighted avg | 0.65 | 0.67 | 0.65 |
| Accuracy | | | 0.67 |

**Table 5.** Multinomial Naïve Bayes classifier Fusion.

| Resulting Tags | Precision | Recall | F1-Score |
|---|---|---|---|
| Bullying | 0.70 | 0.58 | 0.63 |
| Neutral | 0.65 | **0.52** | **0.58** |
| Positive | 0.77 | 0.90 | 0.83 |
| Toxic | 0.30 | 0.37 | 0.33 |
| Macro avg | 0.60 | 0.59 | 0.59 |
| Weighted avg | 0.66 | 0.66 | 0.65 |
| Accuracy | | | 0.66 |

**Table 6.** Support Vector Machines (SVM) Classifier Fusion Metrics.

| Resulting Tags | Precision | Recall | F1-Score |
|---|---|---|---|
| Bullying | 0.62 | 0.77 | 0.69 |
| Neutral | 0.63 | 0.60 | 0.62 |
| Positive | 0.80 | 0.90 | 0.85 |
| Toxic | 0.50 | 0.10 | 0.17 |
| Macro avg | 0.64 | 0.59 | 0.58 |
| Weighted avg | 0.67 | 0.69 | 0.66 |
| Accuracy | | | 0.69 |

In terms of accuracy, the SVM classifier outperforms the other classifiers with 69% followed by RFC with 67%. However, the difference between SVM and RFC is only 2%. In addition, it is about 3% compared to other classifiers (LR and MNB). Indeed, the total accuracy was affected by the accuracy of identifying the comments under the toxic category because it represents the smallest portion of the dataset (13%). Nevertheless, the SVM classifier is characterized by the relatively acceptable precision values (50%) coupled with low recall values, 10%, which results in a low Fl score, 17%. However, this problem can be overcome by using balancing dataset techniques [37,38].

The four classifiers had the best performance metrics (precision, recall, and F1 score) when it came to categorizing positive comments. This is because the number of positive comments represents the largest portion of the dataset (37%). The range of precision was between 76% and 80% with the same value of recall (90%), which results in a high Fl score (82% with LR and MNV, 83% with RFC, and the highest F1 score with SVM was 85%).

The second-best performance metrics were achieved in categorizing the bullying comments. This confirms the previous inference since the bullying comments represent 26% of the dataset. The SVM classifier outperforms the other classifiers with an F1 score of 69% followed by RFC with 67%. However, the difference between SVM and RFC is just 2% and 6% compared to other classifiers (LR and MNB, respectively).

To conclude, the SVM classifier has a considerably higher Fl value than the other classifiers, making it a preferred solution for the problem at hand. Overall, the LR and MNB classifiers achieved the same performance metrics. However, in this paper, we only evaluated the most basic classical classifiers. Many more classifiers and deep learning techniques can be tested and improved upon.

## 5. Conclusions

Due to the negative effects of cyberbullying, recent years have seen an increasing number of studies that collected and annotated datasets related to cyberbullying from social media platforms. However, the majority of the available cyberbullying datasets are in English, with only a few available in Arabic, none of which collected data from

the Instagram platform. Therefore, in this work, we introduce the first Instagram Arabic corpus (sub-class categorization (multi-class)). A total of 46,898 comments were labeled using manual annotation. To consider the dataset as a benchmark, we used SPSS (Kapa statistics) to evaluate the labeling agreements between the three annotators. The result was 0.869 with a *p*-value of $10^{-3}$, indicating near-perfect agreement among the annotators. In addition, we implemented the most basic classifiers (LR, SVM, RFC, and MNB) to evaluate the dataset performance. As a result, the SVM classifier has a considerably higher Fl value than the other classifiers, making it a preferable solution for the problem at hand. The dataset can be used for a range of other NLP tasks, besides cyberbullying classification. This includes dialect identification and sentiment analysis as a training dataset in the context of machine learning. Furthermore, this dataset will help researchers in investigating several preprocessing approaches and machine learning algorithms for developing accurate models for detecting offensive content in online Arabic communications. The complete dataset was publicly released to the research community.

*Future Work*

As for future work, many more classifiers and deep learning techniques can be tested and improved upon. In addition, we can combine both automatic "initial" annotations to detect possible negative/cyberbullying comments and then employ human annotators to just verify those labeling. This is because manual annotation is difficult and time-consuming.

**Author Contributions:** Conceptualization, R.A.; Data curation, S.A. All authors have read and agreed to the published version of the manuscript.

**Funding:** This research received no external funding.

**Institutional Review Board Statement:** The study did not require ethical approval.

**Informed Consent Statement:** Not applicable.

**Data Availability Statement:** The dataset is available at the following link: https://bit.ly/3Md8mj3 (accessed on 20 May 2022).

**Acknowledgments:** This is a part of a project undertaken at the British University in Dubai.

**Conflicts of Interest:** The authors declare no conflict of interest.

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
