# Peer review of "Instagram-Based Benchmark Dataset for Cyberbullying Detection in Arabic Text"

_data, 2022_

Round 1
Reviewer 1 Report
This work is about a study of offensive language on Instagram in Arabic. The study found that the SVM classifier outperformed the other classifiers, with an F1 score of 69% for bullying comments and 85% for positive comments.
The manuscript is innovative as it deals with Arabian and its methods are solid and well described. There are a few aspects that should be enhanced in order to make it published in BDCC.
---
The literature review should be enhanced, particularly when making claims about this dataset being the first of its own for toxic language detection in Arabian. How does this work relates to the authors' work:
https://www.researchgate.net/profile/Osama-Hosam/publication/338454582_Toxic_Comments_Identification_in_Arabic_Social_Media/links/5e15d904299bf10bc39b1ae0/Toxic-Comments-Identification-in-Arabic-Social-Media.pdf
There is a p-value of 0.000 in the Conclusions. Please move that part to the Results and change it to the appropriate value, e.g. < 10^-3. Also, clarify what does it mean to have achieved that value for inter-rater accountability, since that is crucial to dataset re-usability.
The manuscript is well executed but the conclusions lacks key references. Why is detecting cyberbullism so important on Instagram? Stella, Topics Cog Sci 2022 provides a nice review about how online language relates with the mindset of online users, so that detecting cyberbullism online can have key repercussions for detecting cyberbullism in real-world audiences. Similarly, Marzouki et al., Hum Soc. Sci. Comms, 2021, discuss how social media use had a buffering effect over people's anxiety. Might the current dataset be used to investigate whether cyberbullism is linked with increased levels of anxiety in online audiences? Mentioning the above two and a few other concrete scenarios in an extended Discussion would greatly enhance the scope of the manuscript and attract future citations.
Minor points:
Did you obtain any Ethics approval for this study? Were you exempt from it?
Figure 2 has hidden text, please adjust it.
Lines 103-16 - Are those typos expected?
Section 3.5.2 should use more references, even to mainstream books in ML.
Typos: I recommend having the manuscript read and proofed by a native English speaker. Some typos:
interduce -> introduce
Total of -> A total of
Why are some words in Italic in the Conclusions?
The current link provided for open access to the data does not work. Might it be because it is split on two lines? Please consider also uploading the data onto an Open Science platform like the Open Science Foundation: http://osf.io/
Author Response
Dear Reviewers,
We are really very grateful for the feedback and comments that you raised which really assist us in significantly enhancing this work. The productive and valuable remarks enable us to update many parts of the paper as shown by the responses to each comment. Our responses are mentioned below under each comment raised by the reviewer and it is written in (Times New Roman font, blue color). Besides, all the updated parts in the manuscript were highlighted in yellow color in order to be easily tracked by the reviewers

Reviewer 2 Report
The paper is very interesting since the authors focus on cyberbullying in Instagram comments in the Arabic language. Moreover, the authors provide a dataset that would help for further research. It is recommended the authors correct the bibliographic references according to the instructions of the journal they can locate at https://www.mdpi.com/journal/data/instructions. The link to Data Availability Statement is not working. There is a language issue in the paper; polishing is essential. The authors should correct and improve the abstract because it is not written properly. In addition, the authors mention that cyberbullying has social and economic effects; the social effects are obvious, but the economics are not obvious, and the authors should support the argument with related research. The phrase “social media with Abstract” makes no sense in the abstract. The authors in the abstract mention that Instagram is a central social media platform, but they do not mention the research that concluded the above. In the related work section, the footnote 1 h?ps://alt.qcri.org/˜hmubarak/offensive/ is not an active link, in the 2 footnote the at it is not necessary. In the related work section, the authors should add related bibliography in Instagram and cyberbullying, what methodology they use and how the methodology the authors propose differ or is related to the bibliographic review. Figure 1 needs improvement; first, the authors should include the methodology they used to locate accounts; second, the boxes hide the word. Moreover, the authors mention that they crawled the Instagram, but they do not analyze it; it is essential to provide the methodology they used to crawl Instagram if they used an API the code they used. In figure 3 it would be very useful to present the percentages.
Author Response
Dear Reviewers,
We are really very grateful to the feedback and comments that you raised which really assist us to significantly enhance this work. The productive and valuable remarks enable us to update many parts of the paper as shown by the responses to each comment. Our responses are mentioned below under each comment raised by the reviewer and it is written in (Times New Roman font, blue color). Besides, all the updated parts in the manuscript were highlighted in yellow color in order to be easily tracked by the reviewers

Round 2
Reviewer 1 Report
The authors took great care in addressing all my points. The resulting manuscript is way clearer and better scopes.
However, the extensive edits created now an imbalance with too many papers providing statistics and numbers in Section 2 and very little motivation. Please consider additional references that might motivate your approach from a computational social science perspective. I feel my comments about that part were not completely addressed, hence my recommendation for a minor review.
Author Response
The authors are really very grateful for the feedback and comments raised by the reviewer which really assist them in significantly enhancing this work and its presentation. The productive and valuable remarks enable us to update many parts of the paper as shown by the responses to each comment. Our responses are mentioned below under each comment raised by the reviewer and it is written in (Times New Roman font, and blue color). Besides, all the updated parts in the manuscript were highlighted in yellow color in order to be easily tracked by the reviewers.
